# Phase-coherent lightwave communications with frequency combs

Lars Lundberg [1], Mikael Mazur [1], Ali Mirani[1], Benjamin Foo[1], Jochen Schröder[1], Victor Torres-Company [1], Magnus Karlsson [1]*  & Peter A. Andrekson [1]

Fiber-optical networks are a crucial telecommunication infrastructure in society. Wavelength division multiplexing allows for transmitting parallel data streams over the fiber bandwidth, and coherent detection enables the use of sophisticated modulation formats and electronic compensation of signal impairments. Optical frequency combs can replace the multiple lasers used for the different wavelength channels. Beyond multiplexing, it has been suggested that the broadband phase coherence of frequency combs could simplify the receiver scheme by performing joint reception and processing of several wavelength channels, but an experimental validation in a fiber transmission experiment remains elusive. Here we demonstrate and quantify joint reception and processing of several wavelength channels in a full transmission system. We demonstrate two joint processing schemes; one that reduces the phase-tracking complexity and one that increases the transmission performance.

[1] Photonics Laboratory, Department of Microtechnology and Nanoscience, Chalmers University of Technology, Gothenburg, Sweden. *email: magnus.karlsson@chalmers.se

Optical frequency combs were originally conceived for establishing comparisons between atomic clocks[1] and as a tool to synthesize optical frequencies[2,3], but they are also becoming an attractive light source for coherent fiber-optical communications, where they can replace the hundreds of lasers used to carry digital data[4]. One of the key advantages of frequency combs in optical communication is that the separation between consecutive lines is extremely stable. This enables high-spectral-efficiency transmission by minimizing the spectral guard bands between wavelength channels[5,6], and allows for an efficient pre-compensation of fiber nonlinearities[7]. Next in the hierarchy of the comb properties to be exploited is the broadband phase coherence (comb lines are phase locked to each other). This characteristic has been instrumental in expanding the portfolio of comb-based applications[8–10], but its use in lightwave communication systems has been limited.

In a lightwave communication system that uses a frequency comb in place of multiple lasers, the importance of the phase-locking of the comb lines is that all the channels suffer from similar optical phase noise—a fundamental noise source that results in one of the predominant impairments in coherent optical receivers. Phase noise arises mainly from random phase variations of the carrier and local oscillator (LO) light sources, which are usually semiconductor lasers. While the lines of frequency combs also suffer from random phase variations, the broadband phase coherence correlates the variations between WDM channels, as illustrated in Fig. 1a. Modern lightwave communication systems compensate for phase noise using digital signal processing (DSP), which is a main source of power consumption in the receiver. This DSP function can be simplified by using analog methods to phase-lock the LO to the carrier, but at the cost of decreased spectral efficiency and an increased analog complexity[11–13].

When many channels suffer from the same phase noise, the traditional techniques that realize phase tracking on a channel-by-channel basis (Fig. 1c) are redundant. Instead, the channels can be processed jointly, exploiting the phase-noise correlation between them. This is consistent with the superchannel concept, where a set of wavelength-division multiplexed channels are routed and treated as one entity in a transmission network[14]. Having access to multiple channels impaired by the same phase noise means that the phase estimation can be made more efficient in terms of phase tracking capabilities[15–17], or power consumption of the digital electronics[18]. It has been suggested that joint phase processing can be implemented with optical frequency comb sources[19–22] but it has not been demonstrated in a transmission experiment.

In this work, we demonstrate a comb-based transmission system utilizing joint carrier recovery. We transmit 25 channels with 20 GBaud polarization multiplexed 64-ary quadrature amplitude modulation (PM-64QAM) up to 160 km and evaluate joint carrier recovery by receiving two channels simultaneously in synchronized coherent receivers. This is the first demonstration that such joint carrier recovery works in the presence of standard transmission impairments such as chromatic dispersion, polarization drift, amplifier noise and fiber nonlinearities. Our results show that master-slave carrier recovery results in only small penalties, but reduces complexity. Furthermore, we show that a different, joint phase-estimation scheme will increase the tolerance to rapid phase fluctuations induced by nonlinearities in frequency comb-based systems.

## Results

**Frequency comb-based transmission system with joint phase processing.** We consider a multichannel optical transmission system like the one in Fig. 1b, where frequency combs are used as sources for the signal carrier and LO. Then, the phase noise distorting the received signals is correlated between the channels, as illustrated in Fig. 1a. In a traditional signal processing scheme,

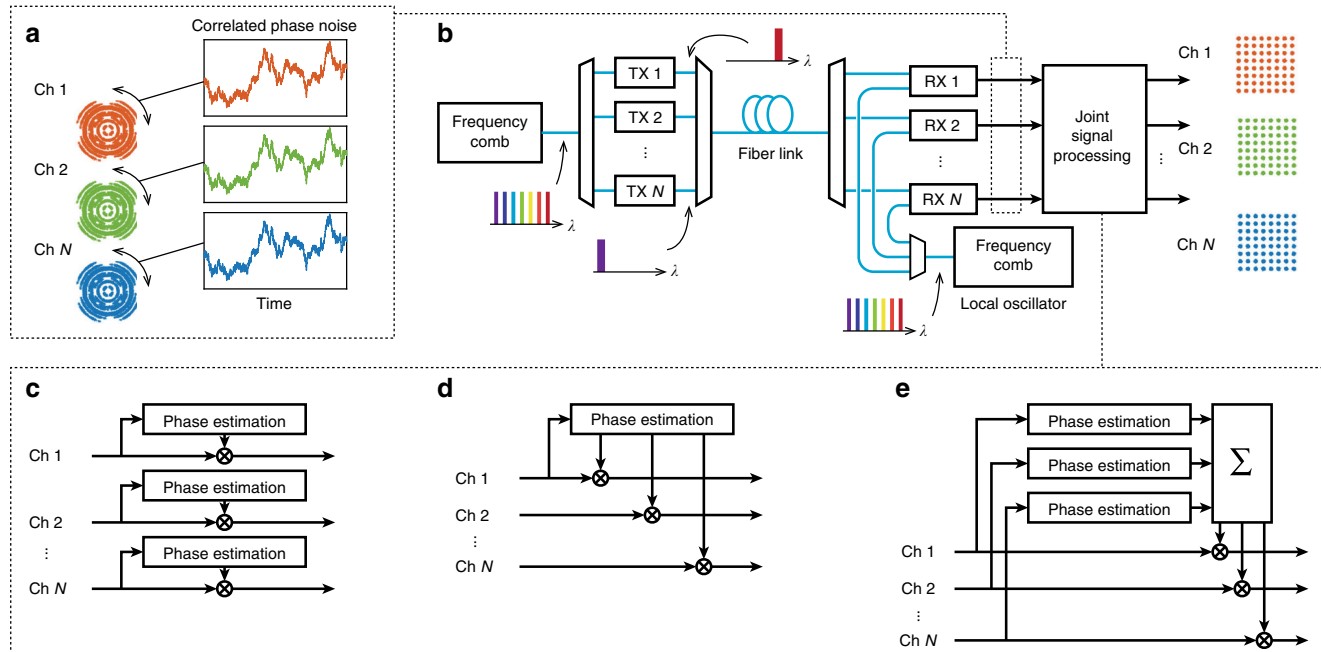

**Fig. 1 Conceptual schematic of phase-coherent communications in wavelength division multiplexing. a** Illustration of the significance of correlated phase noise in a transmission system. The constellation diagrams are distorted by the same phase noise, which enables joint phase processing. **b** The lines of a frequency comb are individually modulated and then transmitted together. In the receiver, a second frequency comb acts as a local oscillator. **c** Traditional carrier recovery with several redundant phase-estimation blocks. **d** Master-slave phase recovery. The phase noise is estimated from one channel and then applied to all channels, eliminating redundant phase-estimation blocks. **e** Joint phase estimation. By averaging the estimated phase noise over several channels, faster phase variations can be detected.

the channels would be processed independently of each other with one phase-estimation block for each channel, as shown in Fig. 1c. However, that scheme does not take advantage of the correlated nature of the phase noise between the comb lines. Instead, the processing can be performed jointly over the channels.

We will study two different schemes for such joint phase processing. The first scheme is best described as master-slave carrier recovery and is based on reusing the phase estimated from one master channel to compensate the phase variations of several slave channels, as illustrated in Fig. 1d. This has the benefit of sharing the relatively demanding phase-estimation block between several channels, thereby reducing the complexity and power consumption of the DSP electronics. The second scheme we propose, illustrated in Fig. 1e, instead uses the multiple versions of the phase noise to improve the phase estimate, which means that the same system can tolerate faster phase variations without incurring penalties. The algorithms are described in more detail in the Methods section.

**Experimental setup.** Our proof-of-principle experiments are based on the joint reception of two WDM channels, originating from an electro-optic frequency comb[23]. In the transmitter (Fig. 2a), all the 25 comb lines are modulated with 8 amplitude levels in each quadrature of both polarizations of the electric field, creating 25 channels with polarization multiplexed 64-ary quadrature amplitude modulation (PM-64QAM) at 20 GBaud spaced 25 GHz apart (giving raw bit-rate of 0.24 Tb/s per channel or 6 Tb/s in the fiber, before coding). The signals are then transmitted through up to two spans of 80 km standard single-mode fiber (SMF) (Fig. 2b). In the receiver (Fig. 2c), two of the channels are jointly received using two synchronized standard coherent receivers. A second, independent, frequency comb acts as a source for the LO. The two frequency combs are seeded from independent continuous wave (CW) lasers and are not synchronized to each other. Since two channels are simultaneously received and recorded, this scheme allows for establishing a quantitative comparison between individual and joint phase tracking. The setup is described in more detail in the Methods section.

**Phase-noise correlation.** In the first set of experiments, we verify that the phase noise remains correlated also after transmission. This can be qualitatively assessed by comparing the phase recovered from the two channels with conventional, independent phase estimation. In Fig. 3a,b the phase curves for the center channel and its neighbor after 80 km transmission are plotted for two launch powers. The curves show high visual similarity, also in

the higher launch power case where nonlinear distortions cause rapid phase fluctuations. The cross-correlation (Fig. 3c) of the two-phase traces confirms the high correlation. The decrease in correlation length at the higher launch power is due to the shorter correlation of the nonlinear phase noise.

**Master-slave performance.** We then study the performance of the master-slave carrier recovery. We quantify the performance by using the generalized mutual information (GMI), which is the maximum data throughput attainable for a bit-wise receiver[24], accounting for the redundancy of an ideal forward-error correction (FEC) code. Today's soft-decision codes can enable throughput quite close to the GMI, which makes it a relevant metric for characterizing the physical channel, in contrast to the bit-error-rate that must assume a specific FEC code. The maximum GMI is modulation format dependent, and in our case of PM-64QAM it is close to 12 bits per four-dimensional symbol (6 bits in each polarization) at the transmitter and is reduced by all signal impairments during propagation through the channel. We compare the performance of traditional independent carrier recovery with joint carrier recovery by comparing the GMI of the same measurement either processed separately, or with the phase information extracted from the other received channel. The impact of frequency separation between the master and the slave was studied by measuring different channel pairs. Since the performance of the channels varied slightly due to power variations of the comb lines, the center channel was always used as the slave channel, while different channels were used as master. The results in Fig. 4a indicate that joint processing can achieve a similar performance to individual processing, in spite of the fact that the phase estimation is only done once. Larger penalties are observed for propagation lengths beyond 80 km for the outermost channels. This is due to a the dispersive walk-off that will decorrelate the channels and to a less extent on the nonlinear phase noise added during propagation in the fiber. Adding a time offset electronically can partly counteract the walk-off and minimize the penalty, but not completely eliminate it. These results indicate that the number of channels that can be jointly processed is limited by the transmission distance, via dispersion and walk-off. This is also confirmed by the simulation results presented in Supplementary Note 1, and a similar limiting effect is also noted in[25] in the context of comb regeneration.

**Joint phase-estimation performance.** We next study the possibility of realizing joint phase estimation (see Fig. 1e). This form of processing is particularly useful to track and compensate the fast phase noise variations that result from the nonlinear interaction

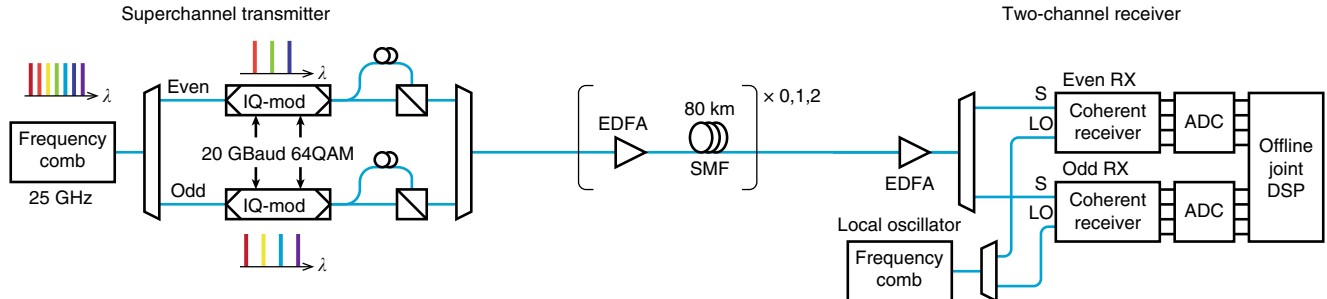

**Fig. 2 Experimental setup for phase-coherent communication.** The lines of an electro-optic frequency comb are divided into even and odd, modulated separately with data and then combined. The signal is transmitted through up to two 80-km fiber spans. In the receiver, one even and one odd channel can be picked to be received simultaneously. A second frequency comb supplies the local oscillator. Digital signal processing (DSP) is performed offline. IQ in-phase quadrature, EDFA erbium-doped fiber amplifier, SMF standard single-mode fiber, S signal, LO local oscillator, RX receiver, ADC analog-to-digital converter.

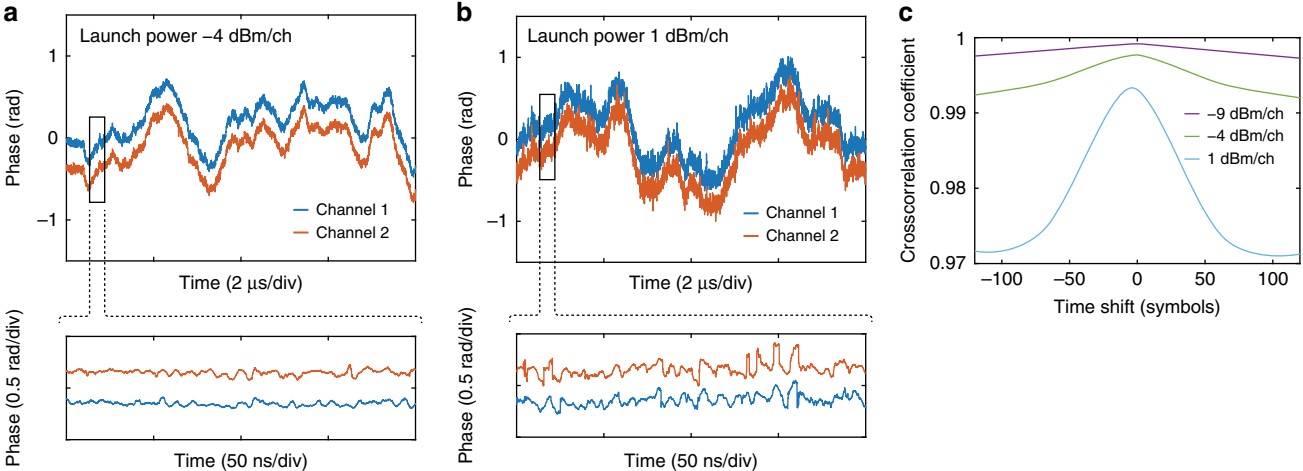

**Fig. 3 Independently recovered phase traces. a** At the optimal launch power (−4 dBm/ch), phase fluctuations are dominated by laser phase noise. **b** At a higher launch power (1 dBm/ch), nonlinear effects cause rapid phase fluctuations. A fixed phase offset has been added to distinguish the two curves. The insets are magnified portions with the linear slopes removed for clarity. **c** The cross-correlation coefficient for the two-phase traces for three launch powers. All plots are for the center channel and its nearest neighbor upon transmission of 80 km.

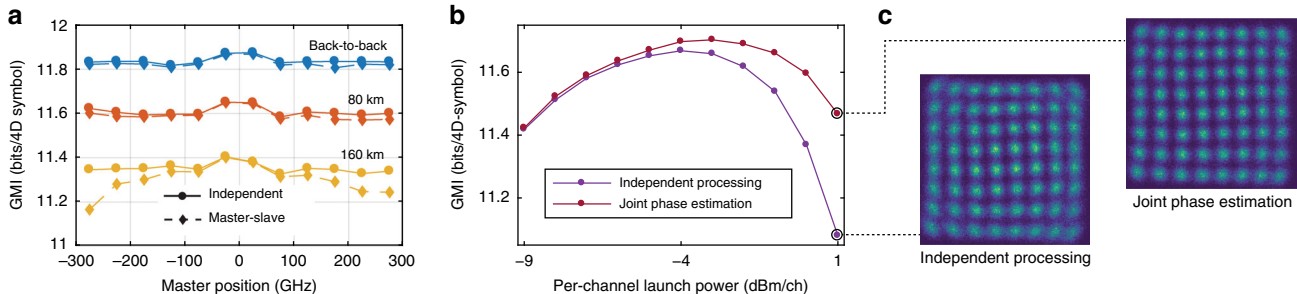

**Fig. 4 Joint phase processing transmission results. a** Performance comparison of master-slave carrier recovery and independent processing of the same measurements. In the measurements, the center channel was used as the slave while the master was varied between different spectral positions. The GMI of the center channel is plotted as a function of the relative spectral position of the master channel. Optimal launch power (−4 dBm/ch) is used. **b** Performance of the center channel as a function of total launch power for joint phase estimation together with its neighbor, compared with independent processing. The transmission distance is 80 km. **c** Comparison of constellation diagrams for independent and joint processing.

in the transmission fiber (Fig. 3b). As before, two neighboring channels in the center of the comb are detected and processed with either conventional single-channel independent phase estimation or joint phase estimation. The same total number of symbols are used for the phase estimation in both cases. In the independent case, all the symbols are taken from one polarization of one wavelength channel, while in the joint case the symbols are distributed over both polarizations of two wavelength channels. This means that the joint scheme uses a four times shorter time averaging, while maintaining the same number of total symbols and the same tolerance to additive noise. The averaging blocks for the phase noise estimation are described in more detail in Supplementary Note 3. At the highest launch power, the performance is improved, and the optimal launch power is increased by 1 dB (Fig. 4b). Constellation diagrams (Fig. 4c) show a noticeable reduction of phase noise in the joint phase-estimation case. The reduced impact of nonlinear phase noise and increased GMI can be translated into an increased data throughput or an increased transmission distance. Even if the nonlinear mitigation at optimum power is modest, our results serve as an illustration of the possible performance increase when strong phase noise is present, as in the highest launch power case. Strong phase noise from other sources could also be more efficiently compensated, with a potential application in systems using a comb source with stronger, but still correlated phase noise.

## Discussion

In summary, we have demonstrated two methods for joint phase processing that utilize the broadband phase coherence of frequency combs for multi-wavelength lightwave communications. This is a fundamental change from the traditional method of treating different wavelength channels independently. Optical frequency combs establish a stable phase relationship between WDM channels, that can be exploited to either significantly reduce receiver complexity, or improve the tracking of more rapid phase fluctuations from, e.g., nonlinear impairments introduced by the fiber link.

The master-slave scheme for phase tracking show a clear potential to reduce complexity, with concomitant power savings in the coherent receiver DSP. For example, in future comb-based superchannels based on four or more wavelength channels, joint processing can use a single phase-tracking subsystem for the entire superchannel, instead of one tracker per wavelength as is needed today.

The joint phase-tracking scheme enables faster response, by reducing the window averaging length needed to suppress the additive noise in phase estimation. This allows for faster phase tracking and tolerance to $2N$ times faster varying phase noise where $N$ is the number of jointly processed wavelength channels. An unexpected benefit is that it also helps mitigating nonlinear phase noise. Since this can be used together with other nonlinear

mitigation schemes such as digital backpropagation, its full potential in nonlinear transmission is yet to be explored, and is a promising field for future research.

Although here demonstrated with single-mode fibers, the scheme could be scaled up thanks to the development of new optical fibers[26–29], allowing to increase the number of jointly processed channels from $1 \times N$ to $L \times N$ by unleashing the space dimension. The optical frequency combs used here are based on benchtop electro-optic comb sources, but the findings are in principle independent of the platform, even though the exact scaling with number of comb lines might differ. For example, the joint processing scheme could benefit from further advances in soliton microcombs and integrated photonics[30]. Together, chip-scale optical frequency combs and joint signal processing have the potential to be a key technology in high-performance, energy-efficient optical transceivers.

## Methods

**Phase relations of detected channels in frequency comb-based systems**. The field of the $n$th line of a frequency comb can be written

$$E_n(t) = |E_n(t)| \exp(j2\pi\nu_0 t + j\phi_0(t) + jn(2\pi ft + \psi(t))), \quad (1)$$

where $\nu_0$ and $\phi_0(t)$ are the center frequency and phase noise of the center line of the comb, $n$ is the line index, $f$ is the frequency spacing and $\psi(t)$ is a phase noise term related to the timing jitter of the comb. This is a general expression valid for different implementations of optical frequency combs[12,31–34].

In a system using frequency combs as signal carrier and LO light sources, the detected signals will have a phase evolution that will be the difference between the phase evolution corresponding to the lines of the carrier and LO frequency combs

$$\phi_n(t) = 2\pi(\nu_{0,S} - \nu_{0,LO})t + (\phi_{0,S}(t) - \phi_{0,LO}(t)) + n(2\pi(f_S - f_{LO})t + \psi_S(t) - \psi_{LO}(t)), \quad (2)$$

where the subscripts $S$ and $LO$ correspond to the signal and LO combs respectively. This equation shows that the channels can be regarded to have the same phase evolution if the combs have sufficiently similar spacing ($f_S \approx f_{LO}$) and the timing jitter noise term is negligible ($\psi_S(t) - \psi_{LO}(t) \approx 0$), forming the basis for joint carrier recovery. The validity of this assumption depends on the exact carrier recovery scheme and the desired number of jointly processed channels. It should be noted that in addition to the intrinsic comb properties, phase variations will also arise from mechanical and thermal disturbances in the transmission fibers. This means that any joint carrier scheme will need to consider some phase differences between the channels, independent of the comb coherence properties. In the description below of our joint carrier-recovery schemes, the practical implementation is described.

Even in the case where the timing-jitter noise or spacing difference is significant, the phase evolution of any channel can be calculated from any two other channels as

$$\phi_k(t) = \phi_n(t) + \frac{k-n}{m-n}[\phi_m(t) - \phi_n(t)]. \quad (3)$$

From a practical perspective, this relation means that a master-slave processing with two master channels could cope with large amounts of timing jitter noise and spacing difference. The allowable spacing difference is however limited by the maximum allowable frequency difference between LO and signal for coherent detection, which is determined by the electrical bandwidth of the photodetectors and electrical components in the receiver.

**Master-slave carrier recovery**. Master-slave carrier recovery relies on using the phase correlations to eliminate redundant carrier recovery blocks, to reduce the complexity and power consumption of the DSP electronics. The basic principle is that the frequency and phase offsets are estimated from one channel (the master channel), and that information is used to compensate the other (slave) channels. The master frequency and phase can be estimated with any algorithm. Specifically, in our method we estimate the frequency offset by finding the peak in the 4th power spectrum[35] and use the blind phase search (BPS) algorithm[16] for phase estimation.

As discussed above, some additional functions are needed to compensate for small frequency and phase differences between the channels. This was implemented as a slow phase tracker to compensate remaining phase variations on the slave channels. As it is desirable to keep any additional processing of the slave channel to a minimum, this slow phase tracking is performed by the adaptive equalizer that is also performing polarization demultiplexing and compensation of polarization mode dispersion. This is achieved by using a decision-directed update algorithm for the equalizer taps, which is sensitive to phase variations, and performing the phase recovery inside the update loop of the equalizer, as is standard for decision-directed algorithms[36]. To realistically evaluate the tracking speed, the equalizer taps were updated every 64th symbol, which emulates

hardware parallellization[37]. This solution could track the timing jitter noise of our combs without penalty, and tolerated up to several tens of kHz of remaining frequency offset. However, the spacing of the transmitter and receiver frequency combs had a difference of around 20 kHz, varying a few kHz over several hours. This spacing difference would hinder joint carrier recovery for any channels but the nearest neighbors due to the scaling with line index. This limitation would not be present in a system with more than two coherent receivers as also the spacing difference could be estimated from the received channels, based on Eq. (3), so in our experiments we measured the frequency difference of the RF clocks and used that information in the signal processing as described in Supplementary Note 2, where we also verify that this approach is valid.

**Joint phase estimation**. Joint phase estimation was performed using the BPS algorithm[16], extended to several channels as described below. The BPS algorithm is based on rotating the received signal with a number of test phase angles, after which the distance to the closest constellation point is calculated for each test phase angle and averaged over several consecutive symbols. The test phase angle with the lowest average distance is chosen as the estimated phase. The averaging is needed to minimize the effect of additive noise on the signal, but the phase tracking speed will be limited by the length of the averaging filter. The optimal length of the averaging filter is a trade-off between tolerance to additive noise and phase tracking speed. A multichannel version of the BPS algorithm extends the averaging to include several channels. Then, a shorter filter length can be used while retaining the same tolerance to additive noise. This is illustrated in Supplementary Note 3, Fig. 7. A more detailed description can be found in[20].

Small phase differences between the channels were compensated in a similar way to the master-slave algorithm, but separated from the main polarization demultiplexing equalizer. Instead, a one-tap decision-directed equalizer was used separately on all channels, with the joint phase estimation taking place inside of the update loop of the equalizers, as illustrated in Supplementary Note 3, Fig. 8.

**Detailed experimental setup**. A schematic of the experimental setup can be seen in Supplementary Note 4, Fig. 9. A frequency comb was created by modulating a CW laser at 1545.32 nm (linewidth <100 kHz) with one phase modulator and one intensity modulator, similar to the comb described in[23]. The modulators were driven by a 25 GHz radio frequency (RF) signal and the comb produced 25 lines with a sufficient power level. The 25-GHz RF-signal was taken from a high-performance fixed-frequency oscillator, with a phase-noise spectrum as plotted in Supplementary Note 4, Fig. 10. The comb was then amplified in an erbium-doped fiber amplifier (EDFA) and fed through an optical processor for flattening. After the optical processor, a 25 GHz interleaver was used to split the comb lines into even and odd. The even and odd lines were then separately modulated with 20 GBaud 64QAM root-raised cosine pulses with a roll-off factor of 0.05 generated with an arbitrary waveform generator operating at 60 GS/s. After modulation, polarization multiplexing was emulated by splitting the signals and delaying one part around 200 symbols before recombining on orthogonal polarizations. The signals were amplified again and recombined using an interleaver. The even and odd paths were length-matched to within 5 symbols on the x-polarization, but the two delays in the polarization multiplexing emulator differed by 10 symbols. The performance was evaluated both in a back-to-back configuration and with up to two 80-km spans of standard single-mode fiber (SMF).

In the receiver setup, two channels could be received simultaneously. The two channels were separated using a multi-port optical processor. The LO lines were taken from a second frequency comb similar to the one in the transmitter. The LO lines were separated using a 25 GHz interleaver. This limited the possible channel combinations to a combination of one even and one odd channel. The LOs were additionally filtered to ensure sufficient extinction ratio. The signals and LOs were mixed in two standard coherent receivers and sampled at 50 GS/s using two synchronized digital sampling oscilloscopes with a bandwidth of 23 GHz.

**Digital signal processing**. The sampled signals were first normalized and ortho-gonalized using the Gram–Schmidt method to compensate for imperfections in the optical hybrid. Then matched filtering and downsampling from 50 GS/s to two samples per symbol (40 GS/s) were performed, followed by dispersion compensation. This was followed by compensation of time skew caused by differences in electrical pathlength of the two receivers. After this, adaptive equalization and carrier recovery were performed. The equalizer had 35 $T_s/2$-spaced taps, where $T_s$ is the symbol time. The taps were pre-converged using the constant modulus algorithm on 400,000 symbols. The output from the pre-convergence was used for coarse frequency offset estimation by raising the signal to 4th power and finding the spectral peak. The frequency estimation was performed using the full 400,000-symbol batch. After pre-convergence, the equalizer was switched to decision-directed mode. Phase estimation and compensation was performed in the update loop of the equalizer, using the BPS algorithm, either independently or jointly as described above. The BPS algorithm used 16 test phases and angle interpolation[38], and the averaging block length was 128 symbols for the master-slave case. For the joint phase-estimation comparison, the averaging block-lengths were 64 for the independent case and 16 for the joint case, further illustrated in Supplementary Note 3, Fig. 7. The equalizer taps were updated every 64th symbol to emulate

hardware parallelization. The step size of the CMA pre-convergence was $10^{-3}$ and the step size of the DD equalizer was $10^{-4}$, for the power-normalized signal. After equalization and carrier recovery, orthogonalization to compensate for modulator bias imperfections was performed. Finally, the performance was evaluated by calculating the GMI using the method in[39], using over 1 million bits.

## Data availability

The data and code that support the findings of this study can be accessed at https://doi.org/10.5281/zenodo.3517781.

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

## Acknowledgements

This work was supported by the Knut and Alice Wallenberg foundation, the Swedish Research Council and the European Research Council (GA 771410). Open access funding provided by Chalmers University of Technology.

## Author contributions

L.L. and M.M. jointly designed the experimental setup and built it with assistance from A.M. L.L. designed the digital processing methods and analyzed the data. B.F. contributed to the interpretation of the nonlinear phase noise results and supplied the code for the split-step simulations. L.L., V.T.C., J.S. and M.K. wrote the paper. M.K., J.S. and P.A.A. supervised the work and provided technical leadership. All co-authors contributed to the paper with critical comments and suggestions.

## Competing interests

The authors declare no competing interests.
