## [Peer Review File · Nature Communications]

Reviewers' comments:

Reviewer #1 (Remarks to the Author):

This is an interesting paper about optical communication with coherent, multiple wavelength sources. The authors describe and characterize methods for phase estimation with a frequency comb that could reduce overall power consumption in high capacity data networks. This area is quite important, supporting technology that is used widely in consumer applications. Use of frequency combs in communications seems to be of realistic importance, therefore the study presented here is likely to capture wide interest. I think this paper should be published in Nature Communications.

The paper really excels in terms of describing the phase estimation schemes and experimental results. The ideas presented here are concrete and explored comprehensively.

My primary complaint about this paper is that it reads in between a research paper and a review paper. The text presents various eventualities of future systems that it seems may or may not be important. In my mind the question is where is the real opportunity here, and how does this paper put one in a position to realize it. Some revisions to clarify how the results of the paper should be used would help significantly.

I wonder about this phrasing: "Fundamentally, joint phase processing presented here will be limited by the phase coherence of the comb sources, which have a line dependent phase noise term [25] that will cause a phase noise difference that increases with the separation between the channels [20]. This term is dependent on the comb generation technique." (Emphasie mine.) In one sense, the comb phase noise is independent of generation technique because phase noise scaling in combs is fundamental. However, the type of comb could change the phase noise and its fixed point. Therefore, the communication scheme is likely dependent on the comb used, and the authors perhaps should consider these possibilities in more detail. I do not feel this point is already addressed in the literature.

"Optical frequency combs establish a fully coherent relationship between WDM channels..." Here in the final summary, the phrase fully coherent confuses me with regard to the challenges with coherence.

Since the comb coherence is critical, it would be helpful to specify the electronic details of the EO combs used in the work.

Reviewer #2 (Remarks to the Author):

The paper describes two applications of joint phase processing based on the use of optical frequency combs in coherent optical fiber communication systems: a master-slave phase estimation configuration and a joint phase estimation configuration. Both configurations have been considered previously, including papers by the authors (e.g., [20]). Thus, the contribution of the paper resides in the specific implementations and the experimental results. The results are new, but quite limited in scope. The paper would benefit from additional experimental results.

Comments and suggestions

- 1) The Abstract refers to two novel signal processing schemes, yet does not explicitly indicate what they are. Given that the two schemes (master-slave phase estimation configuration and a joint phase estimation configuration) are known, what is the rationale for describing them as novel? (Definition of novel (on-line Merriam-Webster dictionary): new and not resembling something formerly known or used.)
- 2) The last sentence of the Abstract does not directly relate to the content of the paper and is quite vague in referring to unspecified advances in SDM and photonic integration. It does not seem appropriate for an Abstract.
- 3) Figs. 1(a) and 1(c) are cited parenthetically in the Introduction without any supporting descriptive or explanatory information, either there or elsewhere. Perhaps, it is best to present and

describe them in Section 2. Fig. 1(b) does not appear to be mentioned at all in the body of the paper.

4) The paper claims "This approach allows for a more efficient estimation and compensation of optical phase noise in coherent communication systems, which can significantly simplify the signal processing or increase the transmission performance." The quantitative results to support this statement are quite limited in scope: Fig. 3 present results for the phase noise correlation, Fig. 4(a) presents results for the GMI for the master-slave phase estimation configuration, and Fig. 4(b) present results for the GMI for the joint phase estimation configuration with only two channels. Fig. 4(a) indicates the simplification in the signal processing is accompanied by performance penalties of up to 0.2 bits/4D symbol. Fig. 4(b) indicates an increase in the GMI of about 0.04 bits/4D symbol at a launch power of about 11 dBm. Since the decrease in GMI of up to 0.2 bits/4D symbol is described as a slight penalty, what adjective would appropriately describe the increase in GMI of 0.04 bits/4D symbol in Fig. 4(b)? The specific fiber length used to obtain the results in Fig. 4(b) does not appear to be indicated. A description of the superchannel (abscissa label) for which the results in Fig. 4(b) were obtained is not provided. As written, it is difficult to properly interpret the results.

5) The proposed approaches require that the receiver signal processing for a multiple number of channels be very strictly time-synchronized. Current commercial implementations preclude this. An envisioned receiver architecture that is capable of capturing and synchronously processing a multiple number of channels needs to be described, as does the extent of the difficulties in achieving this with state-of-the-art technologies. How would the requirement for synchronization impact the overall simplification in the signal processing? The additional complexity that arises from having fewer phase estimation blocks is not addressed.

6) Are the specified launch powers in Fig. 4 for a single channel or for 25 channels? The caption for Fig. 3 states that the total launch power is indicated, and while it could be assumed that this also applies elsewhere, a clear statement would be helpful.

7) Figs. 2 and 4 use the word superchannel but this term does not appear in the text. Given the description of the experimental setup, the use of this term requires clarification.

8) Section 2.2. "... all the comb lines are modulated ... creating 25 channels..." Were just 25 comb lines generated?

9) The zoomed-in results in Figs. 3(a) and 3(b) do not have the same slopes as the figures above from which they are extracted. An explanation is needed.

10) Parameter values for all signal processing algorithms should be included.

11) Section 2.3. "The curves show high visual similarity, also in the higher launch power case where nonlinear distortions cause rapid phase fluctuations." Does the similarity of the nonlinear distortion in the higher launch power case change if the two channels are far apart, rather than being adjacent?

12) Section 2.4. Change "... is reduced by all signal impairment" to "... is reduced by all signal impairments"

13) Section 2.4. Change "... making GMI a good characteristic of the physical channel" to "... making GMI a good metric for characterizing the physical channel".

14) Section 3. Change "... are likely to limit the optimal number of jointly channels" to "... are likely to limit the optimal number of jointly processed channels".

15) Section 4.2. Change "The basic principle is that the frequency and phase is estimated ..." to "The basic principle is that the frequency and phase are estimated ..."

16) Section 4.4. Change "... differed 10 symbols" to "... differed by 10 symbols".

17) Provide more detail about the estimation procedure for supplementary note 1, including the frequency resolution. Explicitly indicate what is meant by the channel frequency difference (abscissa). Comment on the expectations for different fiber lengths.

Reviewer #3 (Remarks to the Author):

The manuscript titled "Phase-coherent lightwave communications with frequency combs" reports two signal processing schemes for a more efficient estimation and compensation of optical phase noise in coherent communication systems using optical frequency combs. The manuscript is well written and technically sound. However, I do not think that the presented work has broad

readership or enough significance to be published in Nature Communications. I would recommend the authors to submit this manuscript to more specific journals in optical communications.

Reviewer #1 (Remarks to the Author):

This is an interesting paper about optical communication with coherent, multiple wavelength sources. The authors describe and characterize methods for phase estimation with a frequency comb that could reduce overall power consumption in high capacity data networks. This area is quite important, supporting technology that is used widely in consumer applications. Use of frequency combs in communications seems to be of realistic importance, therefore the study presented here is likely to capture wide interest. I think this paper should be published in Nature Communications.

The paper really excels in terms of describing the phase estimation schemes and experimental results. The ideas presented here are concrete and explored comprehensively.

My primary complaint about this paper is that it reads in between a research paper and a review paper. The text presents various eventualities of future systems that it seems may or may not be important. In my mind the question is where is the real opportunity here, and how does this paper put one in a position to realize it. Some revisions to clarify how the results of the paper should be used would help significantly.

We agree that the real opportunities for system improvements were somewhat unclear, but the purpose of this paper is to describe the potential and pros and cons for joint processing. In the revision we have made changes to the discussion section, so that it now has a clearer conclusion and a

discussion of the potential impacts. The discussion on impairments has been extended and moved to the supplementary material as note 1.

I wonder about this phrasing: “Fundamentally, joint phase processing presented here will be limited by the phase coherence of the comb sources, which have a line dependent phase noise term [25] that will cause a phase noise difference that increases with the separation between the channels [20]. This term is dependent on the comb generation technique.” (Emphasis mine.) In one sense, the comb phase noise is independent of generation technique because phase noise scaling in combs is fundamental. However, the type of comb could change the phase noise and its fixed point. Therefore, the communication scheme is likely dependent on the comb used, and the authors perhaps should consider these possibilities in more detail. I do not feel this point is already addressed in the literature.

Indeed, different comb generation techniques have different phase-noise properties, and the fixed point of the phase noise is not necessarily in the center of the comb spectrum. Nevertheless, frequency combs do follow Eq. (1) (Section 4.1), which implies that comb lines close to each other have correlated phase noise, and that the full comb phase properties can be determined from only two lines (Eq. (3)). Additionally, the benefits of our schemes can be shown already for 2 or more jointly processed lines, but the scaling to more lines will of course depend on the comb technology used and their line coherence. Not all comb generation techniques will result in combs with as good coherence properties, but to discuss the coherence properties of various comb technologies is an active research topic in itself, and we believe, beyond the scope of this work.

Nonetheless, we have added a clause on the impact on different comb technologies in sec 3.

“Optical frequency combs establish a fully coherent relationship between WDM channels...” Here in the final summary, the phrase fully coherent confuses me with regard to the challenges with coherence.

We have reformulated this to say “stable phase relationship”.

Since the comb coherence is critical, it would be helpful to specify the electronic details of the EO combs used in the work.

A short description on the EO-combs used and phase-noise spectrum of an oscillator used for one of the combs has been added in Supplementary note 4.

Reviewer #2 (Remarks to the Author):

The paper describes two applications of joint phase processing based on the use of optical frequency combs in coherent optical fiber communication systems: a master-slave phase estimation configuration and a joint phase estimation configuration. Both configurations have been considered previously, including papers by the authors (e.g., [20]). Thus, the contribution of the paper resides in the specific implementations and the experimental results. The results are new, but quite limited in scope. The paper would benefit from additional experimental results.

Comments and suggestions

1) The Abstract refers to two novel signal processing schemes, yet does not explicitly indicate what they are. Given that the two schemes (master-slave phase estimation configuration and a joint phase estimation configuration) are known, what is the rationale for describing them as novel? (Definition of novel (on-line Merriam-Webster dictionary): new and not resembling something formerly known or used.)

We understand the reviewer's concern about the wording in the abstract, and we are aware that the DSP schemes have been previously described. We have rewritten parts of the Abstract and introduction to emphasize that the novelty of this work lies in the full demonstration of the system concept including transmission. Although similar concepts have been described previously, they have not considered transmission, which means that they have not considered effects from amplifier noise, dispersive walk-off or fiber nonlinearities. Especially the possibility to counteract the impact of nonlinear phase is something that had not been predicted and discussed before.

2) The last sentence of the Abstract does not directly relate to the content of the paper and is quite vague in referring to unspecified advances in SDM and photonic integration. It does not seem appropriate for an Abstract.

The Abstract has been partly rewritten and this text removed.

3) Figs. 1(a) and 1(c) are cited parenthetically in the Introduction without any supporting descriptive or explanatory information, either there or elsewhere. Perhaps, it is best to present and describe them in Section 2. Fig. 1(b) does not appear to be mentioned at all in the body of the paper.

We have added a paragraph describing the system concept in Section 2, where Fig. 1 is referred to and fully described.

4) The paper claims "This approach allows for a more efficient estimation and compensation of optical phase noise in coherent communication systems, which can significantly simplify the signal processing or increase the transmission performance." The quantitative results to support this statement are quite limited in scope: Fig. 3 present results for the phase noise correlation, Fig. 4(a) presents results for the GMI for the master-slave phase estimation configuration, and Fig. 4(b) present results for the GMI for the joint phase estimation configuration with only two channels. Fig. 4(a) indicates the simplification in the signal processing is accompanied by performance penalties of up to 0.2 bits/4D symbol. Fig. 4(b)

indicates an increase in the GMI of about 0.04 bits/4D symbol at a launch power of about 11 dBm. Since the decrease in GMI of up to 0.2 bits/4D symbol is described as a slight penalty, what adjective would appropriately describe the increase in GMI of 0.04 bits/4D symbol in Fig. 4(b)?

We understand the reviewers concern about the description of our quantitative results and have therefore reformulated the section describing the master-slave results to discuss the combined penalty of transmission distance and frequency spacing between master and slave. We have also added simulation results in the supplementary materials to provide a more comprehensive view on these limitations. Indeed, the worst-case penalty for master-slave processing in the 180-km transmission case is 0.2 bits, but only for the largest tested frequency spacing. For spacings up to 225GHz the penalty is below 0.1 bits. This corresponds to 19 jointly processed channels, which already represents a highly interesting application.

We have added simulations as a supplementary note 1 to further quantify how the interplay of dispersion and nonlinearity affects the penalties in the master-slave case, as well as text in the end of sec 2.4 to discuss this.

We agree that the performance increase is relatively modest for joint phase estimation when comparing at the optimal launch powers. However, if comparing for the highest launch power where the nonlinear phase noise is stronger, the performance increase is more significant. While the optimal launch power is the relevant comparison point when considering nonlinearity compensation, the comparisons at higher launch powers can be seen as representing a more general situation with strong phase noise that is not necessarily originating from nonlinear effects. In other words, our results illustrate the possibility of achieving a performance increase in systems that are affected by other sources of strong, but correlated phase noise. Interestingly, this source of correlated phase noise could arise from the light source itself. As a result, our joint phase estimation method would allow the use of certain frequency comb technologies which typically have a phase noise too strong to allow them to be used in coherent communications. This is now mentioned at the end of sec 2.5 and in the discussion section of the manuscript.

We have also extended the discussion on the phase and frequency estimation in the supplementary note 2.

The specific fiber length used to obtain the results in Fig. 4(b) does not appear to be indicated. A description of the superchannel (abscissa label) for which the results in Fig. 4(b) were obtained is not provided. As written, it is difficult to properly interpret the results.

The fiber length was 80km and the superchannel setup was the same as used in all experimental results in the paper, i.e. 25 times 25GHz-space 20GBaud PM-64QAM. This has been clarified in the revised manuscript.

5) The proposed approaches require that the receiver signal processing for a multiple number of channels be very strictly time-synchronized. Current commercial implementations preclude this. An envisioned receiver architecture that is capable of capturing and synchronously processing a multiple number of channels needs to be described, as does the extent of the difficulties in achieving this with state-of-the-art technologies. How would the requirement for synchronization impact the overall simplification in the signal processing? The additional complexity that arises from having fewer phase estimation blocks is not addressed.

We agree with the reviewer that the timing synchronization issues are important to consider, so we have added an analysis (Supplementary note 1, Fig 3) of the sensitivity to timing offsets between the

channels, which shows that penalty is below around 0.1 bit for over 100 symbols of timing offset for both 80km and 160 km transmission. We also identify dispersive walk-off as the main limiting factor from these simulations (see Supplementary note 1), which is now mentioned last in sec 2.5.

Even if a specific electronic implementation of the proposed algorithms will require different electronic hardware, we do not see this as a fundamental limitation. On the contrary, we see a trend for using the superchannel concept, as single-channel throughput demand increases faster than the available symbol rates. We would also like to point out that e.g. Infinera has commercial multi-channel coherent transceivers, where one ASIC processes two channels, and although this does not necessarily mean that the DSP is joint, or synchronized over multiple channels, it points out that this is a likely direction of development.

6) Are the specified launch powers in Fig. 4 for a single channel or for 25 channels? The caption for Fig. 3 states that the total launch power is indicated, and while it could be assumed that this also applies elsewhere, a clear statement would be helpful.

In the previous version of the manuscript, all launch powers were given for all 25 channels, but we have changed this to average per-channel launch powers, since this is a more general measure. We have added clarifying notes about this in all relevant places.

7) Figs. 2 and 4 use the word superchannel but this term does not appear in the text. Given the description of the experimental setup, the use of this term requires clarification.

We added a description and a citation to Chandrasekhar's original proposal of superchannels in the introduction section.

8) Section 2.2. "... all the comb lines are modulated ... creating 25 channels..." Were just 25 comb lines generated?

Yes, just 25 usable comb lines were generated, i.e. 25 comb lines with sufficient power. This has been clarified in the manuscript in sec. 4.4.

9) The zoomed-in results in Figs. 3(a) and 3(b) do not have the same slopes as the figures above from which they are extracted. An explanation is needed.

The difference in the slopes is because the slopes have been removed from the zoomed-in results to produce a clearer plot. The purpose is to visualize the similarity in the phase variations. This has now been clarified in the figure caption.

10) Parameter values for all signal processing algorithms should be included.

Additional parameter values have been given in sec. 4.5.

11) Section 2.3. "The curves show high visual similarity, also in the higher launch power case where nonlinear distortions cause rapid phase fluctuations." Does the similarity of the nonlinear

distortion in the higher launch power case change if the two channels are far apart, rather than being adjacent?

We have studied the correlation of the nonlinear phase noise using numerical simulations, the results of which are presented in Supplementary note 1, Fig. 4. The figure shows that the nonlinear phase noise is indeed correlated also for larger line separations, apart from a time offset which is caused by dispersive walk-off.

12) Section 2.4. Change “... is reduced by all signal impairment” to “... is reduced by all signal impairments”

13) Section 2.4. Change “... making GMI a good characteristic of the physical channel” to “... making GMI a good metric for characterizing the physical channel”.

14) Section 3. Change “... are likely to limit the optimal number of jointly channels” to “... are likely to limit the optimal number of jointly processed channels”.

15) Section 4.2. Change “The basic principle is that the frequency and phase is estimated ...” to “The basic principle is that the frequency and phase are estimated ...”

16) Section 4.4. Change “... differed 10 symbols” to “... differed by 10 symbols”.

These language issues have been corrected. We thank the reviewer for spotting them.

17) Provide more detail about the estimation procedure for supplementary note 1, including the frequency resolution. Explicitly indicate what is meant by the channel frequency difference (abscissa). Comment on the expectations for different fiber lengths.

A more detailed description has been added and is now supplementary note 2, which has been significantly extended.

Reviewer #3 (Remarks to the Author):

The manuscript titled "Phase-coherent lightwave communications with frequency combs" reports two signal processing schemes for a more efficient estimation and compensation of optical phase noise in coherent communication systems using optical frequency combs. The manuscript is well written and technically sound. However, I do not think that the presented work has broad readership or enough significance to be published in Nature Communications. I would recommend the authors to submit this manuscript to more specific journals in optical communications.

We thank the reviewer for taking the time to read and provide feedback on our manuscript, but we respectfully disagree with the statement that the significance is not enough to be published in Nature Communications, and note the support of our point of view from referee 1. The use of frequency combs in optical communication is believed to be a promising route even if the comb is only considered a compact and stable multiwavelength light source, and some impressive previous work on this topic has received broad interest (e.g. [4]).

The significance of our work is that we, in contrast to previous work, demonstrate that frequency combs have further-reaching benefits beyond just replacing an array of lasers. This is important since it shows that the use of frequency combs may influence the design of the whole system, including the signal processing schemes. We take advantage of the phase-coherence of frequency combs, which has been a crucial enabler for other applications, but which has not been exploited in optical data transmission before in this direct way. Thus, our work connects, in a new way, a broad range of areas, ranging from laser physics, through communication-system design, to signal processing, for which we believe it motivates publication in Nature Communications.

REVIEWERS' COMMENTS:

Reviewer #1 (Remarks to the Author):

I feel the authors appropriately addressed my original comments. I am in favor of publication.

Reviewer #2 (Remarks to the Author):

The authors have added a considerable amount of new material and effectively addressed the reviewers' comments.

Minor comments and suggestions

- 1) Section 1. This is the first demonstration that such joint carrier recovery works in **the** presence of standard transmission impairments
- 2) Section 3. Considering "... while in the joint case the symbols are distributed over both polarizations of two wavelength channels. This means that the joint scheme uses a four times shorter time averaging..." and "This allows for faster phase tracking and tolerance to N times faster varying phase noise where N is the number of jointly processed channels", should N be $2N$?
- 3) Section 4. ... where ν_0 and $\phi_0(t)$ **is are** the center frequency and phase noise
- 4) Section 4. "is however limited by the maximum allowable difference between LO and signal..." What property of the LO and signal is difference referring to?
- 5) Section 4.5. "matched filtering and downsampling ... **was were** performed."
- 6) Section 4.5. "adaptive equalization and carrier recovery ... **was were** performed."
- 7) Supp. Note 1. The note reports simulation results. The captions for Figs. 1 and 2 refer to simulation results. The caption for Fig. 3 refers to experimental results. Are these simulation results?
- 8) Supp. Note 1. "Although our experiments investigate a situation corresponding to several tens of jointly processed channels, a more likely first step would involve only a couple of wavelengths." This statement is confusing in that it refers to experiments in a note on simulations (see above comment) and it refers to several tens of jointly processed channels whereas the paper considers 25 channels.
- 9) Supp. Note 2. "We evaluated the estimated frequency difference for different frequency spacing between the two channels" Are the frequency spacings between the two channels $\pm 25, \pm 75, \pm 125, \pm 175, \pm 225$ and ± 275 GHz? If so, perhaps this could be indicated.
- 10) Supp. Note 2. For channel frequency differences of ± 25 GHz, the estimated frequency difference noticeably changes as the fiber length increases. The authors offer an explanation for the dependence on the channel frequency difference for each of the three cases (back-to-back, 80 km, and 160 km). Can an explanation be provided for the dependence on the fiber length?

Reviewer #3 (Remarks to the Author):

Thank you for the extensive revision of the original manuscript. The revised manuscript clearly describes the novelty and importance of their work, the first demonstration and analysis of joint processing schemes utilizing the phase coherence of frequency combs. There is no doubt that frequency-comb-based optical communication systems are important and getting more attention in recent years. However, these systems have been studied over a decade and I am not sure if demonstrating known DSP schemes for incremental improvement on GMI can justify the publication in Nature Communications. As an aside, the significance of ref [4] lies on the fact that chip-integrable combs can revolutionize the optical communication systems and their work demonstrated very high transmission capacity. So, I would like to emphasize that my only concern is the limited scope and significance, and there is no doubt of the quality of work. I cannot strongly recommend its publication in Nature Communications, but because my opinion on the scope and significance can be a subjective one, it is OK to publish if the other reviewers recommend.

Response to referee comments, and final edits:

1) Section 1. This is the first demonstration that such joint carrier recovery works in the presence of standard transmission impairments

1) Done

2) Section 3. Considering "... while in the joint case the symbols are distributed over both polarizations of two wavelength channels. This means that the joint scheme uses a four times shorter time averaging..." and "This allows for faster phase tracking and tolerance to N times faster varying phase noise where N is the number of jointly processed channels", should N be 2N?

2) Yes and we also changed to "where N is the number of jointly processed wavelength channels" for clarity,

3) Section 4. ... where ω_0 and $\phi_0(t)$ is are the center frequency and phase noise

3) Done

4) Section 4. "is however limited by the maximum allowable difference between LO and signal..." What property of the LO and signal is difference referring to?

4) We clarified this sentence to "...however limited by the maximum allowable frequency difference between LO and signal for coherent detection, which is determined by the electrical bandwidth of the photodetectors and electrical components in the receiver."

5) Section 4.5. "matched filtering and downsampling ... was were performed."

5) Done.

6) Section 4.5. "adaptive equalization and carrier recovery ... was were performed."

6) Done.

7) Supp. Note 1. The note reports simulation results. The captions for Figs. 1 and 2 refer to simulation results. The caption for Fig. 3 refers to experimental results. Are these simulation results?

7) We clarified the section heading to “Transmission simulations and impairments from delays” which should indicate that not all is simulations.

8) Supp. Note 1. “Although our experiments investigate a situation corresponding to several tens of jointly processed channels, a more likely first step would involve only a couple of wavelengths.” This statement is confusing in that it refers to experiments in a note on simulations (see above comment) and it refers to several tens of jointly processed channels whereas the paper considers 25 channels.

8) We added the following sentence to emphasize that Fig 3 is not a simulation but an experimental check requested by a referee. “In addition to the above simulations, we performed an additional experimental check on the tolerable channel skew. With today’s...”

We also changed the quoted sentence to “Although our experiments investigate a situation corresponding to several tens of jointly processed channels, a more likely first step in an online DSP implementation would involve only a couple of wavelengths.”

9) Supp. Note 2. “We evaluated the estimated frequency difference for different frequency spacing between the two channels” Are the frequency spacings between the two channels ± 25 , ± 75 , ± 125 , ± 175 , ± 225 and ± 275 GHz? If so, perhaps this could be indicated.

9) We added the following sentence: “We used the odd channel set, located at $\pm (25, 75, 125, 175, 225, 275)$ GHz. “

10) Supp. Note 2. For channel frequency differences of ± 25 GHz, the estimated frequency difference noticeably changes as the fiber length increases. The authors offer an explanation for the dependence on the channel frequency difference for each of the three cases (back-to-back, 80 km, and 160 km). Can an explanation be provided for the dependence on the fiber length?

10) We have no good explanation for this, but added this sentence “The reason for the reduced variation in the estimates at longer distances is not totally clear to us, but is related to the reduced SNR. “

All these instances are marked with red in the revised manuscript.